# Warm Season Turfgrass Equine Sports Surfaces: An Experimental Comparison of the Independence of Simple Measurements Used for Surface Characterization

**DOI:** 10.3390/ani13050811

**Published:** 2023-02-23

**Authors:** María Alejandra Blanco, Facundo Nicolas Di Rado, Michael (Mick) Peterson

**Affiliations:** 1Racing Surfaces Testing Laboratory, Lexington, KY 40502, USA; 2Faculty of Engineering and Agricultural Sciences, Pontifical Catholic University of Argentina, Buenos Aires 1300, Argentina; 3School of Engineering, Agricultural and Food Sciences, University of Moron, Moron 1708, Argentina; 4Biosystems and Agricultural Engineering, College of Engineering, University of Kentucky, Lexington, KY 40503, USA

**Keywords:** equine, arenas, turfgrass, base layers, portable tools, safety, equine welfare

## Abstract

**Simple Summary:**

Turfgrass equine sports surfaces represent only a subset of surfaces used in equine sports but are often used in some of the most high-profile events. The use of grass adds complexity to the management because of the interaction of horseshoe and foot with the surface. The orientation of the roots allows the toe to penetrate the surface while at the same time reinforcing the surface during propulsion. The availability of injury data and the sensitivity of the public to rider and horse safety make safety-related research a primary focus for horse racing. The epidemiological research associated with risk due to surface conditions is generally based on ratings on the turf, typically a qualitative judgment from officials. The focus of this paper is on the use of several commercially available or easily constructed portable instruments which could be used for quantitative ratings. The results suggest that testing conditions may determine the data consistency. As in previous work, caution must be exercised in the interpretation of the results since these tools have not been demonstrated to correlate to either performance or safety of the surface.

**Abstract:**

Turfgrass in equine sports has clear advantages over other types of reinforcement but adds complexity to the management. This study investigates factors that influence the turfgrass’ surface performance and how the use of a drainage package and a geotextile reinforcement affect quantitative measurements of turfgrass. The measurements are made using affordable, lightweight testing tools that are readily available or easily constructed. Eight boxes with turfgrass over a mix of the arena with peat at a consistent depth were tested for volumetric moisture content (VMC %) with time–domain reflectometry (TDR), the rotational peak shear device (RPS), the impact test device (ITD), soil cone penetrometer (SCP), and the Going Stick (GS). Results obtained using TDR, RPS, ITD, SCP, and GS indicate that the presence of the geotextile and drainage package was mainly detected by VMC (%), SCP detected geotextile addition, and GS detected the interaction of geotextile × drainage package. Linear regression showed SCP and GS are related to geotextile and was positively correlated between them and negatively with VMC (%). The testing showed some limitations of these devices, mainly related to moisture content and sod composition, but the potential exists to utilize these devices for quality control as well as for the monitoring of maintenance of the surfaces when controlling the range of both VMC (%) and sod constitution.

## 1. Introduction

For most equine sports surfaces, the competition surfaces are composed of sand, sand reinforced with fiber, or turfgrass [1]. The use of turfgrass as an equine sports surface adds complexity to the interaction of horseshoe and foot with the surface. Turf allows the hoof and shoe to penetrate into the surface while the root system provides additional reinforcement to support the hoof to avoid a shear failure, or divot, in the surface [2,3] Turf also has clear advantages over other types of surfaces like sand stabilized with fibers, since, unlike added fibers, the damaged roots will regrow. Furthermore, the roots tend to be preferentially oriented in a vertical direction which increases shear strength but does not inhibit vertical penetration. The vertical root orientation allows the toe to penetrate the surface between the roots while, at the same time, the roots reinforce the surface during propulsion when large horizontal forces are applied. As in other athletic sports fields root- zones are constructed using a high proportion of sand to improve growth and performance [4]. However, this can cause a lack of cohesion and a poor support to the horseshoe, so stabilization of the root zone is needed. The stabilization of the surface can be achieved by reinforcement of the root zone with synthetic materials [4,5,6]. 

Recent research has considered the effect of the type of surface and condition on the incidence of injuries and performance [7,8,9]. Horse racing has a primary focus on safety. The availability of injury data and the sensitivity of the public to rider and horse safety are an opportunity to keep the social license on horseracing [10,11]. The epidemiological research associated with risk due to surface conditions is generally based on ratings on the turf. In general, the ratings of surface conditions are qualitative judgments from racing officials [12]. Quantitative measurements may be obtained from the track, but the publicly available ratings are adjusted according to the experience of racing official and may not match the quantitative measurements. A notable exception is the data from New Zealand which is strictly quantitative [13,14]. New Zealand uses the Longchamp Penetrometer to quantify the surface effect on race times, but the same data has also been recently shown to be useful in quantifying risk associated with the surface condition [15]. Ratings in Australia, Japan, and the United Kingdom are a combination of qualitative and quantitative data which are interpreted by racing officials that have been shown to predict risk in some cases, although results are mixed [16,17,18]. Existing tools used for handicapping and measurement of the performance of the turf surface may provide a promising basis for research. Currently, available track ratings may not be comparable between racing jurisdictions or even within a single racing jurisdiction because of qualitative adjustment of the quantitative measurements. 

In other equine sports like showjumping or dressage, the effect of condition or rating of the surface on injuries is less readily available since both surface ratings and injury data are less consistently available. However, it is generally accepted that understanding the risk to the horse (and rider) may require an improved understanding of the condition of equestrian turf surfaces [1,19,20]. 

The focus of this paper was on use of several commercially available or easily constructed portable instruments which are used in equine sports to test surface conditions. Since surface moisture content is closely related to surface quality a moisture probe based on an older ASTM standard D6565 was included in the work to test volumetric moisture content (VMC %). The rotational peak shear (RPS), based on ASTM F2333, was used in a previous study [19,20,21] to test sand surfaces with variable results. In turfgrasses a similar device described by Canaway and Bell [22] was used successfully to differentiate species and cultivars [23], reinforcement materials in the root zone [24,25,26], and biomass [27], but it was less sensitive to soil condition resulting of traffic and compaction [28,29,30]. The impact test device (ITD) based on ASTM D5874-16 is similar to the Clegg hammer, the Impact Test Index (ITI) express the deformation of the surface after dropping. Guertal and Han [31] found that Clegg hammer could differentiate between traffic and non-traffic areas in turf tracks but could not distinguish different levels of traffic. Two commonly used measurement devices, the Longchamp penetrometer and the Going Stick, were also considered. While no testing standard exists for the Longchamp penetrometer, it is the one device in current use [18,32] which is supported by data that correlates the measurements to the performance of the horse through a general descriptive model [13,14] as well as being correlated with injury [15]. A similar tool, the dynamic cone penetrometer, is used instead of the Longchamp penetrometer since it is already included as a standard with only slight modifications from an international standard [33]. The Going Stick is commonly used in some countries, and while it is currently not a standard test and is only provided through a commercial entity, it was included in the current form. The objective of this investigation is to identify if these tools provide independent measures of the turfgrass surface condition. Based on the outcome, the tools required to characterize the surface would be able for use in future large scale epidemiological research.

## 2. Materials and Methods

### 2.1. General Overview

The effects of two turfgrass building systems, which consisted of a drainage package (Dr) above the limestone base and the addition of geotextile chips (G) as reinforcement of the root zone, were tested with five portable devices. The surface was prepared with and without a drainage package over the limestone to establish if there was an effect that would modify the mechanical properties of the surface. The drainage package was a three-layer system consisting of two layers of geotextile with a geomesh layer between the layers of geotextile. Since the profile was built with sand substrate geotextile chips were added to stabilize or reinforce the profile.

### 2.2. Study Design

The design was a randomized block with two factors (2^3^) with two-level with two repetitions (eight boxes). The factors were the addition of geotextile to the sand, which constitutes a depth cushion and the inclusion of a drainage layer over the limestone base. The absence of geotextile was designated as G1, with the inclusion of 2 $\frac{kg}{m^{2}}$ of geotextile chips mixed through the profile, designated as G2. In both cases, the depth cushion is 10 cm deep over the base. The base was a compacted limestone with 0.7% of cross-slope either directly under the depth cushion, designated as Dr1, or a drainage package was placed between the depth cushion and the limestone base, designated as Dr2 (Figure 1). The drainage package consisted of two layers of geotextile fabric with a geomesh layer placed between them. Testing was performed on two dates: 3 March 2018 and 30 March 2018. The two dates were considered as blocks in the analysis. Gravimetric moisture content was determined in the laboratory for both dates treatments on 3 March 2018; it was 44.6% ± 9.40, and on 30 March 2018 was 47.6% ± 8.94 [34].

The testing box dimensions were 1 m × 1 m with an overall prepared depth of 0.20 m. Boxes were constructed as described in a previously published study [21] with the dimensions established from prior testing [19,35]. The 10 cm of sand over the 30 cm base material was applied in two layers. Each 5 cm layer was compacted separately using a 4 kg mass dropped three times from a height of 0.30 m onto an area of 0.20 m by 0.17 m for all treatments. Above the sand, sphagnum moss peat and Bermudagrass Tifway 419 sods were placed. All boxes were fertilized with controlled-release complex fertilizers, which were fully coated and with slow solubility. The fertilizer applied was a mixture of nitrogen, phosphorous, and potassium (NPK) with 30 g/m^2^ of Basacote (Compo Expert^®^ GmbH, Münster, Germany) [36] and 50 g/m^2^ of Floranid permanent (Compo Expert^®^ GmbH, Münster, Germany) [37]. Irrigation was supplied during the whole experiment. The mowing height was kept to 30–50 mm in every box and across the experiment.

The growing medium was 92.3% sand, 2.6% silt, and 5.1% clay (Testing by Racing Surfaces Testing Laboratory, Lexington, Kentucky) [38]. The 2 cm sod was not of the same composition as the profile and was acquired from a turfgrass producer. The geotextile was 100% polyester (Lab Cor Materials, LLC, Seattle, WA, USA) [39]. The sand was 90.6% quartz, 2.7% potassium feldspar, 2.5% plagioclase, 0.2% calcite, and 4% phyllosilicates (KT Geoservices, Gunnison, CO, USA). The Geomesh was high-density polyethylene. The materials characterization, sand, and geotextile used are the same as a prior study on arena surfaces without turfgrass [21]. Because of the living nature of turfgrass and also as a soil pore system after the experiment on 1 April 2018, samples of depth cushion from both geotextile and drainage package treatments were taken for water–air availability testing (Laboratory of Substrates of the Faculty of Agronomy of University of Buenos Aires) in accordance with European standards CEN EN-3041 [40,41], results are in Appendix A.

### 2.3. In Situ Measurement of Test Boxes

All testing was performed on both testing dates using five in situ measurement tools. The five measurement tools were: a volumetric moisture content tester (VMC) consistent with ASTM D6780 [42], a rotational traction tester, which was used to measure the rotational peak shear (RPS) based on ASTM F2333 [43], an impact test device which measured the impact test index (ITI) based on ASTM D5874, [44], a dynamic penetrometer also called the soil cone penetrometer (SCP) consistent with ASTM D 6951-03 [33] and the Going Stick, a commercial instrument which measures the Going Stick index (GSI). To the extent possible, these devices were used in accordance with all applicable standards. While no current standard exists for the Going Stick, the use and design was consistent with other published literature [21,45,46]. Three repetitions of all measurements were achieved during the first date, and five on the second date (Table 1).

The field moisture probe (Spectrum Field Scout TDR-100, Aurora, IL, USA) using time-domain reflectometry was used to measure the volumetric moisture content (VMC %). The probe was equipped with two measuring rods of 8 cm in length.

The rotational peak shear (RPS) was recorded using a torsional shear tester based on a modification of ASTM F2333-04 [43]. The standard device was modified by eliminating the cleats on the disk and replacing them with a horseshoe. The horseshoe included two 2.5 cm long cleats. The rotational peak shear (RPS) load was measured with a digital torque wrench with a range of 4–200 NM and precision of 0.08 Nm (Model ARM602-4, ACDelco, Detroit, MI, USA). In order to minimize operator variability, the same person, shown in Figure 2, performed all of the testing, which was consistent with best practice [47].

The surface hardness and resistance to compaction was measured using an impact test device (ITD) based on ASTM D5874-16. The deformation of the surface in cm is used to describe the surface and will be referred to as the Impact Test Index (ITI).

The soil cone penetrometer (SCP) testing was based on ASTM D6951-03 [33]. The SCP test uses repeated drops of a mass onto a rod to drive a cone-shaped penetrator into the soil a fixed distance [48]. The test method is used with a number of different configurations, which depend both on the application and on the soil type. While the current testing was a typical agricultural application, a smaller mass dropped was dropped at a shorter distance to account for the finite depth of the test box and the sandy soil [49]. A hammer with a mass of 2 kg was dropped from a height of 50 cm to the impact surface on the anvil. The number of drops required to press the end of the rod to a depth of 100 mm is measured. The end of the rod was a right circular cone with a 20 mm diameter base and 15 mm height, and the shaft mass was 3.5 kg. While previous research had correlated this apparatus to both the shear strength of the soil and the California Bearing Ratio [48], these relationships would depend on the exact configuration of the instrument.

The Going Stick measures the penetration resistance and the resistance to rotation of the blade in the turf [45]. The GSI is an integrated proxy of the two measured values. The GSI is commonly used in Thoroughbred racing to describe the surface conditions in a number of jurisdictions [45].

### 2.4. Statistical Analysis

For all the data obtained, the analysis of variance was performed using commercial statistical analysis software (Infostat version 2, Buenos Aires, Argentina). For comparison of marginal means the *t*-test was performed. Values of *p* < 0.05 were considered statistically significant. Linear regression analysis between dependent variables and independent variables and Pearson’s coefficients of correlation were calculated to identify the degree of association between dependent variables.

The proposed model for the testing of the boxes is:$$Y_{ijk}=µ+{Dr}_{i}+G_{j}+B_{k}+\left({DrG}_{ij}\right)+\left(DrB_{ik}\right)+(GB_{jk)}+{\left(DrGB_{i\dot{J}k}\right)+e}_{ijk}$$

*Dr*: Drainage _(1,2)_*G*: Geotextile _(1,2)_*B* _(1,2)_: Blocks.

## 3. Results

### 3.1. In Situ Measurement of Test Boxes

All measurements obtained from the five devices showed to be significant for some of the treatments, drainage (Dr), and the addition of geotextile (G) and Blocks (Dates). Results of variables tested for each treatment (Table 2 and Table 3). Significant effects are shown through f and *p* values (Table 3).

Linear regression analysis (Table 4) demonstrated that VMC (%) was positively associated with Dr and G (R^2^ = 0.54), SCP was associated with G and GSI was associated with G. Correlations between devices (Table 5) showed that VMC (%) was negative correlated with penetrometer (*p* < 0.0216; *p* = −0.29) and GSI was positive correlated with SCP (*p* < 0.0025; *p* = 0.37).

### 3.2. Moisture Probe

The volumetric moisture content (VMC %) was significant for triple interaction block × drainage × geotextile (Figure 3). The block effect showed higher values of VMC on date 2. When geotextile was present, VMC % was the lowest on both dates. Drainage packages preserved higher VMC % when geotextile was not present on both dates. Regression analysis identified VMC was significantly associated (R^2^ = 0.54) with Dr (*p* < 0.001) and with G (*p* < 0.001) (Table 4). Correlation between devices showed that VMC (%) was negative correlated with SCP (r = −0.29, *p* < 0.0216).

### 3.3. Rotational Peak Shear

The rotational peak shear (RPS) was not significant for both treatments, but it was significant to block effect, which means that the moment of testing showed a different result (B1 (3 March 2018): 44.41 Nm ± 5.56 and B2 (30 March 2018): 47.64 ± 5.49; respectively *p* < 0.0274) (Table 2).

### 3.4. Impact Test Device

The measurements made with the impact test device (ITD) were not statistically significant for all two factors and blocks (Table 2).

### 3.5. Cone Penetration

The soil cone penetrometer (SCP) was significant for geotextile (*p* < 0.0118) (Table 2). Geotextile addition had the highest mean values of SCP (G1 = 2.78 ± 0.75 and G2 = 3.38 ± 0.94). When regression was run SCP, R^2^ was 0.17 and was significantly associated with G (*p* < 0.0061) (Table 4).

### 3.6. Going Stick

GSI was statistically significant for drainage package (Dr) (Table 2 and Table 3). GSI showed higher values during the second date (B2) and with the drainage package as well as being significant for the interaction between the drainage package and geotextile addition (Dr × G) (Figure 4 and Table 3). The treatments B_2_Dr_2_G_2_ (6.57 ± 2.65) and B_2_Dr_2_G_1_ (5.53 ± 0.88) showed higher GS index values. When linear regression was run, GSI was associated with geotextile (Table 4). Correlation between devices showed that GSI was correlated with SCP (*p* < 0.0025 r = 0.37) (Table 5).

## 4. Discussion

To be useful, the basic tools being used in this study should be sensitive to the treatments of the surfaces in the boxes that are known to have an impact on the performance of an equestrian turf surface, such as drainage and geotextile [1]. The insensitivity of some of the measurements to the treatments is at least as notable as the sensitivity of other measurements.

Moisture content is generally understood as one of the most important factors in the dynamic properties of equestrian surfaces [19,50]. What is most notable about the results from the VMC (%) measure is that the treatments of geotextile and the drainage package are most clearly identified simply by monitoring the moisture content (Table 2). The dynamic of this effect seems to be influenced by pluviometry, soil temperature (B1 = 23 mm/month and average temperature 26 °C; B2 = 35 mm/month and average temperature 22.1 °C), and evapotranspiration occurred during both periods (end of summer). Drainage and geotextile combined seem to reduce VMC (%), while drainage seems to reduce the flow of water (Table A1, Figure A1). Looking at the interactions, the most notable conclusion is that VMC (%) is sensitive to nearly all the effects that are known to influence performance.

The measurement of RPS was notable for the insensitivity to the treatments and lacking repeatability between test days. The RPS was previously showed significant differences in a sand–geotextile mix [21]. However, in this study, the profile includes a warm-season turfgrass, Bermuda Hybrid “Tifway 419”, which has a high shoot density and horizontal stems. In sand surfaces, the effect of geotextile added on the response of the surface is related to the relative motion between the horseshoe and the surface. This resistance results in higher forces during a pivoting movement closely related to the RPS measurement [51]. The turfgrass might modify the relative motion between the horseshoe and the surface, such that the turf, and not the presence of geotextile, dominates the response. The magnitude of the shear readings in the current experiment were 20 points above the previous testing in sand. The differences between dates (B1 = 44.41 Nm ± 5.56; B2 = 47.64 Nm ± 5.49) were detected, but if the turf itself and not the profile dominates the response, then the additional growth of the horizontal stems could account for the differences between days. This should be a useful measurement and would represent a characteristic of turfgrass arenas of warm-grasses which may be important for performance. Bermuda Hybrid “Tifway 419” has been reported as an intermediate traction species (42.9 Nm) [52] in comparison with the other two warm grasses.

Both the ITI and SCP do not appear to provide any additional information from the test boxes. Not only was the ITI insensitive to the two treatments it also did not correlate to any of the other measurement tools (Table 5).

In contrast to our previous work, ITD was not sensitive to any factor in this experiment. Like with the RPS, the additional components of turfgrass profile, such as horizontal stem dispositions, shoot density, leaves thickness, and roots, may interfere with the displacement of a low weight of the drop; thus, mowing height may be obscuring drainage and geotextile effects. Other authors have found vertical displacement with a Clegg Hammer of 5.8 mm for Bermuda “Tifway 419” [52] in field capacity at a mowing height of 15 mm; in this experiment, the range of vertical displacement get from ITD was from 8 to 13 mm at a mowing height of 30–50 mm and a higher VMC (%). The mowing height chosen here is more related to the racetrack than other sports like polo playgrounds. The same authors [52] also found a negative correlation between dry matter and vertical deformation in football pitches. Our preliminary data on the partition of the dry matter (shoot/root) showed that turfgrass growing with drainage package or geotextile improves partitioning to shoot, these conditions and the addition of high moisture content achieved in the study may have operated on the lack of sensitivity shown by ITD. Some authors found a well-defined exponential trend of decreasing compressive strength with increasing soil saturation [53] regardless of the soil type. The lack of sensitivity of ITD may be a combination of decreased compressive strength with increased volumetric moisture and turfgrass biomass that change the configuration.

SCP showed higher values when geotextile is present (Table 2). SCP, as well as GSI, have also shown sensitivity to date of testing but detecting the effect of Dr and G. Bermuda Hybrid “Tifway 419” has also been reported as a high penetration resistance in soil playground surfaces [52], although in this experiment resistance to penetration recorded by SCP was an average of 3.08 drops (min = 1 and max:6), which means a lower value in comparison with other studies, and may be related to the high VMC (%), regarding present correlations as variance between VMC (%) and SCP explained (*p* < 0.0216, r = −0.29). This result is in accordance with the decreasing of the values of the dynamic cone penetrometer as a CBR% as saturation increased [53].

The GSI was sensitive to different aspects of surface design. GSI was shown to consistently detect the presence of the geotextile when the drainage package was also present. The surface with geotextile had higher GSI, and alternative when drainage package is present. In both moments of testing, the GSI value was in a range equivalent as a good to soft surface rating, with lower values in the first testing (B1) and higher values in the second testing moment (B2). Although a correlation exists between the GSI and SCP results, the GSI includes a combination of longitudinal shear and penetration resistance. Because of the effect of turf, both properties were captured by the GSI under the conditions included in this study.

The correlation shown between devices suggests that geotextile is modifying the penetration resistance force. Both results would be compatible with firmness and cushioning surface, although they don’t measure peak load.

The drainage package may be acting as a subsurface water storage, as shown by the water availability test (Figure A1). This is consistent with the results of McInnes and Thomas, where the drainage package on top of limestone reduced the water flux rate and may also provide a more consistent interface [54]. This means that the reduction in water flux promotes a special arrangement of particle distribution, even though all of the treatments had the same size initial particle distribution. This finding is relevant to turfgrass health, although, for the experimental conditions considered, VMC’s (%) was high.

## 5. Conclusions

Many turfgrasses equine sports surfaces have different components than sand surfaces. However, in this experiment, the composition of the topsoil is closely related to a sand surface. The turfgrasses canopy adds another dimension to the interaction of the hoof–surface. The effect and the interaction of drainage and geotextile were tested with five devices. Previous studies on sand boxes were confirmed with the effect of drainage package and geotextile addition able to be detected with some of the simple instruments used in this experiment. In this study, TDR proved to be the most reliable tool, able to detect differences promoted by geotextile or drainage package despite the high moisture content. The effect of drainage on surface properties is likely to be related to the presence of both a vertical and horizontal water movement and the subsequent effect on packing particles that lead to higher VMC at higher water tension values and, in this sense, adding consistency. As in our previous study, the ability to test functional properties of the surface may be limited, however, due to the lower loads and lower load rates from these instruments. In addition, the turfgrasses canopy adds complexity through sward structure and variations in the height of mowing. These turfgrass properties contribute to the complex surface-hoof features and limit the ability to detect differences with simpler devices such as RPS, ITD, SCP, or GS. The different sensitivity of some tools like RPS, SPC, and GSI between dates highlights the interest in changes in the growth rate of shoots and roots and thickness of leaves regarding the season and affecting hoof-surface interaction. The lack of sensitivity of some devices may also be influenced by the method of turfgrass establishment. The turf was applied as sod, which included a 2 cm thick layer of silty clay soil which may be a barrier to the detection of drainage and geotextile treatments by the RPS and ITD. While future research may include washing the sod, this is not typically done in most installations. Moisture also plays a role in the lack of sensitivity of some devices. However, since the arenas or racetracks are outdoors, some combination of factors like drainage package and high volumetric moisture content may lead to turf with an off condition. Understanding the consequences of injury risk is the primary objective of the use of these tools.

Future work should consider alternative approaches that are also suited for monitoring the proper maintenance of the turfgrass surfaces. Methods such as vertical cutting and aeration are critical to ensuring that a high-quality surface can be provided that results in a consistent performance over time.

## Figures and Tables

**Figure 1 animals-13-00811-f001:**
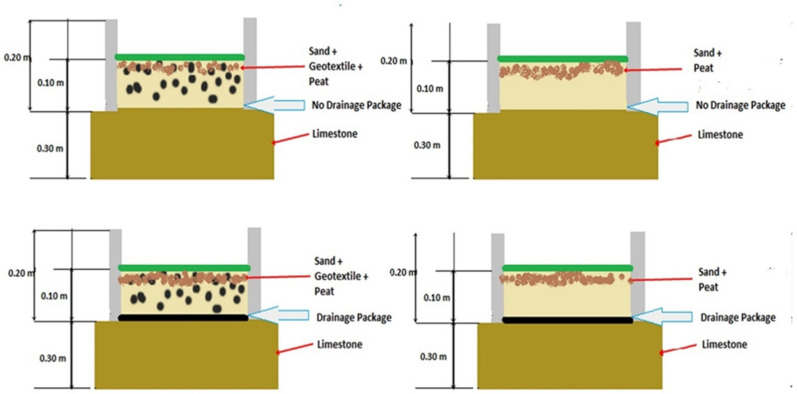
Scheme of boxes design, including the depth of the top cushion material consisting of sand (G1), or sand with geotextile (G2 = 2 $\frac{kg}{m^{2}}$) over limestone (Dr1), or over drainage package (Dr2).

**Figure 2 animals-13-00811-f002:**
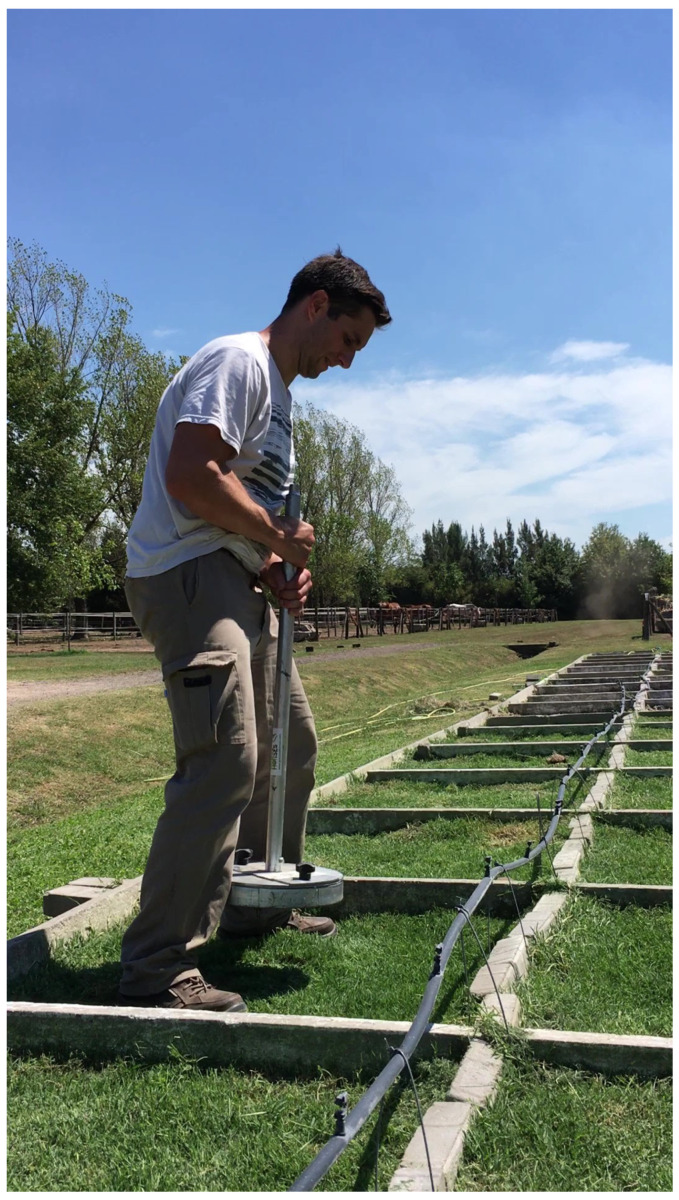
One of the members of the team stood over the boxes of turfgrasses at the moment of the RPS testing (3 March 2018), ©2018 Maria Alejandra Blanco.

**Figure 3 animals-13-00811-f003:**
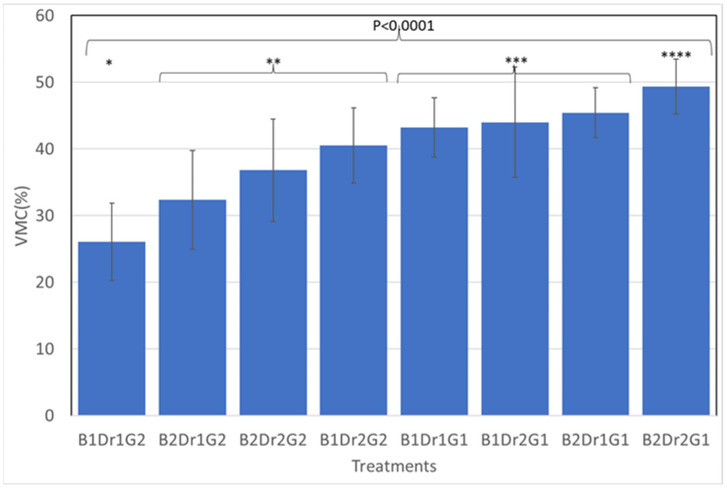
The mean of triple interaction blocks × drainage × geotextile for volumetric moisture content (VMC %) B1: 3 March; B2: 30 March; Dr1: without drainage package, Dr2: with drainage package; G1: without geotextile; G2: with 2 kg/m^2^ of geotextile. The stars indicate significant statistical differences (*). (Numbers of asteriks *, **, ***, **** indicates *p*-value).

**Figure 4 animals-13-00811-f004:**
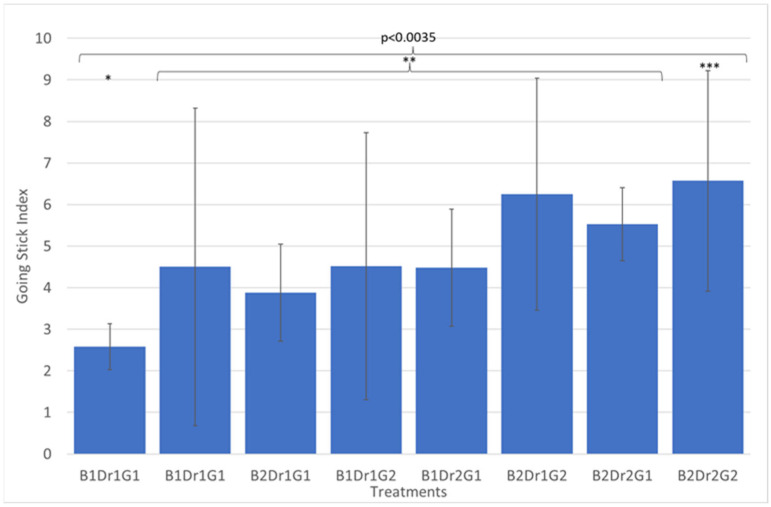
The mean of triple interaction block × drainage × geotextile for Going Stick Index (GSI) B1: 3 March; B2: 30 March; Dr1: without drainage package, Dr2: with drainage package; G1: without geotextile; G2: with 2 kg/m^2^ of geotextile. The stars indicate significant statistical differences (*). (Numbers of asteriks *, **, *** indicates *p*-value).

**Table 1 animals-13-00811-t001:** Replications of treatments and variables from tools used in the experiment on both two dates: 3 March and 30 March 2018.

Replications		Drainage Package (Dr = 2 Repetitions)		Geotextile (G = 2 Repetitions)		Drainage Package × Geotextile	
Variables	Dates	3 March 2018	30 March 2018	3 March 2018	30 March 2018	3 March 2018	30 March 2018
**VMC**	2	3	5	3	5	3	5
**RPS**	2	3	5	3	5	3	5
**ITI**	2	3	5	3	5	3	5
**SCP**	2	3	5	3	5	3	5
**GSI**	2	3	5	3	5	3	5

**Table 2 animals-13-00811-t002:** Statistics for the VMC (%), RPS, ITI, SCP, and GSI and GSS (f and *p*-value), to three factors drainage, geotextile addition, and blocks. ANOVA parametric variables *p* < 0.05 statistic test f; ^1^ ANOVA for non-parametric variables: Kruskal–Wallis Test *p* < 0.05 statistic test H.

Variable	Drainage (Dr)		Geotextile (G)		Blocks (B)	
	f (H)	*p*	f (H)	*p*	f (H)	*p*
**VMC (%) ^1^**	4.94	0.0262 *	27.92	0.0001 *	1.46	0.2275
**RPS**	2.46	0.1221	1.38	0.2445	5.13	0.0275 *
**ITI**	1.01	0.3193	0.07	0.7975	0.22	0.6377
**SCP ^1^**	1.01	0.287	5.52	0.0118 *	2.50	0.094
**GSI ^1^**	4.79	0.0286 *	2.16	0.1413	10.04	0.0015 *

Tukey test Alpha = 0.05 *p* < 0.05. (*) significant difference.

**Table 3 animals-13-00811-t003:** f (H) and *p*-values of interactions between factors: drainage × geotextile, drainage × blocks, geotextile × blocks, and drainage × geotextile × blocks over the five variables (VMC, RPS, ITI, SCP, and GSI) ANOVA *p* < 0.05 statistic test f; ^1^ ANOVA for non-parametric variables: Kruskal–Wallis Test *p* < 0.05 statistic test H.

Variable	Dr × G		Dr × B		G × B		Dr × G × B	
	f (H)	*p*	f (H)	*p*	f (H)	*p*	f (H)	*p*
**VMC ^1^ (%)**	33.75	0.0001 *	6.75	0.082	30.32	0.0001 *	37.87	0.0001 *
**RPS**	0.04	0.8500	0.13	0.7161	0.99	0.3240	0.72	0.4001
**ITI**	0.57	0.4539	1.03	0.3152	0.16	0.6825	0.13	0.7237
**SCP ^1^**	6.76	0.055	3.73	0.2325	8.35	0.0225 *	9.72	0.132
**GSI ^1^**	10.46	0.015 *	14.84	0.0020 *	12.78	0.0054 *	21.19	0.0035 *

Tukey test Alpha = 0.05 *p* < 0.05. (*) significant difference.

**Table 4 animals-13-00811-t004:** Regression coefficients (R^2^), and linear model coefficients for every variable VMC (%), RPS, ITI, SCP, and GSI. Significant values at *p* < 0.05 (*).

Variables	R^2^	Linear Model Coefficients			
		Constant	Dr	G	B
**VMC (%)**	0.54	45.55 *	5.46 *	−11.87 *	2.52
**RPS**	0.13	35.49 *	2.16	1.63	3.23 *
**ITI**	0.02	0.02 *	−0.0013	0.0003	−0.00006
**SCP**	0.17	1.67 *	0.30	0.82 *	0.54
**GSI**	0.22	−1.12	0.97	1.43 *	1.54 *

**Table 5 animals-13-00811-t005:** Pearson coefficients correlations r and *p* for every variable VMC (%), RPS, ITI, SCP, and GSI on treatments. (*) significant difference.

Variables	VMC (%)		RPS		ITI		SCP		GSI	
	r	*p*	r	*p*	r	*p*	r	*p*	r	*p*
**VMC (%)**	1	0.001	−0.07	0.6083	0.07	0.5654	−0.29	0.0216 *	−0.15	0.2439
**RPS**	−0.07	0.6083	1	0.001	−0.01	0.9451	0.05	0.6807	0.17	0.1746
**ITI**	0.07	0.5654	−0.01	0.9451	1	0.001	−0.19	0.1418	−0.09	0.4655
**SCP**	−0.29	0.0216 *	0.05	0.6807	−0.19	0.1418	1	0.001	0.37	0.0025 *
**GSI**	−0.15	0.2439	0.17	0.1746	−0.09	0.4655	0.37	0.0025 *	1	0.001

* Significant values at *p* < 0.05.

## Data Availability

Not applicable.

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
