# Peer review of "Warm Season Turfgrass Equine Sports Surfaces: An Experimental Comparison of the Independence of Simple Measurements Used for Surface Characterization"

_animals, 2023, doi:10.3390/ani13050811_

Round 1

Reviewer 1 Report

This is a timely and interesting article

I only have a few comments that relate to grammar and clarity around the sward height of the Bermuda grass and consistency of sward height across testing events.

Specific comments

Throughout manuscript – please check the manuscript for use of past tense.  Ideally past tense should be used when describing what was performed and the results found.  Examples of this are the use if “is” rather than “was”.

Line 24 not sure reinforcement is the appropriate word here – but not I don’t have a suitable suggestion coming to mind – possibly considering restructuring the sentence which would add greater clarity.

Materials and methods

Somewhere in here I think there needs to be mention of the sward height and if this was consistent across the testing dates and boxes.  This is important as it is mentioned in discussion as a confounder.

Results

Line 246 shown – rather than showed

Line 251 replace exhibited with demonstrated

Table 4 please include the coefficients for the model rather than just if coefficient were significant or not

Line 263-264 consider rewriting sentence …Regression analysis identified VMC was significantly associated (R2 =0.52) with Dr (p<0.001) and G (p<0.001) – no need to report p values to so many decimal places – once p<0.001 then only need to report to p<0.001.

Line 335 in contrast to our previous…..

Line 351  consider …..compressive strength with increased volumetric moisture and turfagrass biomass that changed the configuration.

Line 359  present correlations as variance explained and then significance …eg r=0.29, p=0.022

Line 362  GSI provide to be a consistent or robust estimator …

Line 371 ….storage as the water availability….

Line 372 In agreement with McInnes and Thomas [64], …..

References

Please check the formatting of the authors names or how they are entered in your endnote or reference package – the application of the MDPI output style appears to have incorrectly muddled up authors and their initials eg reference 8, 29 and 40 as examples that were very obvious.

Author Response

Regards

Reviewer 2 Report

The focus of this manuscript is on the quantification of turf conditions. This is relevant because turf conditions are frequently included as variables in studies of the epidemiology of injury.

Introduction.

The term “reinforcement” is used extensively. It would be useful to define specifically what is meant by the term in the context of the manuscript.

It would be helpful to more clearly differentiate between equestrian sports that are regulated by the FEI and horse racing. FEI standards are not applicable to horseracing jurisdictions.

It is hard to understand how the countries listed as utilising turf as the major racing surface were selected. In Australia Thoroughbred racing is conducted almost universally on turf surfaces. The 2019 IFHA annual report, the most recent available, lists Australia as having the second largest number of annual race meetings behind the USA. The relevance of the paragraph describing international differences is not obvious to me.

The discussion of the use of track ratings as variables in the assessment of performance is also of questionable relevance.

Overall the introduction fails to provide a sensible basis for the study and should be rewritten. It would be sufficient to say that

  1. turf conditions may influence the risk of injury in racing horses,
  2. turf conditions are measured using a variety of techniques which vary from jurisdiction to jurisdiction,
  3. understanding differences in these techniques will be important if epidemiological data is to be shared between jurisdictions, and the aim of this study was to improve understanding of the different measurement techniques.

Methods.

The first paragraph of 2.1 Study Design does not make sense where it is currently positioned in the manuscript. A more general overview of the study design is required.

2.2 Includes the sentence  “(a)ll testing was performed on both testing dates using five in-situ measurement tools in the second date”. What does this sentence mean - were measurements made on both dates or only the 2nd date?

The meaning of terms such as “volumetric moisture content”, “rotational peak shear” and “impact test index” must be defined somewhere-it would be appropriate to define these variables in the introduction in place of the largely irrelevant content.

The statistical analysis is appropriate.

Results.

The captions for the tables should be improved in order to make them more descriptive of the information contained in the table.

Discussion.

It is difficult to make sense of the discussion as it is written.

General.

It seems that the information contained in the manuscript is of value, but it is poorly written and is unsuitable for publication at this time.

Several of the references in the Bibliography are incomplete and all references should be checked to ensure they comply with the requirements of the journal.

Author Response

Regards

Round 2

Reviewer 2 Report

There is still some confusion in terminology. The terms “equestrian” and “sports horses” are used in the introduction but the focus seems to be entirely on horse racing. The other terms are not usually used in relation to racing. The focus of the introduction requires further clarification.
